# The Rapidly Expanding Group of *RB1*-Deleted Soft Tissue Tumors: An Updated Review

**DOI:** 10.3390/diagnostics11030430

**Published:** 2021-03-03

**Authors:** Sasha Libbrecht, Jo Van Dorpe, David Creytens

**Affiliations:** 1Department of Pathology, Ghent University Hospital, Ghent University, 9000 Ghent, Belgium; sasha.libbrecht@gmail.com (S.L.); jo.vandorpe@uzgent.be (J.V.D.); 2CRIG, Cancer Research Institute Ghent, Ghent University Hospital, Ghent University, 9000 Ghent, Belgium

**Keywords:** retinoblastoma, soft tissue tumor, RB1, spindle cell/pleomorphic lipoma, atypical spindle cell/pleomorphic lipomatous tumor, pleomorphic liposarcoma, myofibroblastoma, cellular angiofibroma, acral fibromyxoma

## Abstract

The classification of soft tissue tumors has evolved considerably in the last decade, largely due to advances in understanding the pathogenetic basis of many of these, sometimes rare, tumors. Deletion of Retinoblastoma 1 (*RB1*), a well-known tumor suppressor gene, has been implicated in the tumorigenesis of a particular group of soft tissue neoplasms. This group of so-called “*RB1*-deleted soft tissue tumors” has been rapidly expanding in recent years, currently consisting of spindle cell/pleomorphic lipoma, atypical spindle cell/pleomorphic lipomatous tumor, pleomorphic liposarcoma, myofibroblastoma, cellular angiofibroma, and acral fibromyxoma. Most of these neoplasms, except pleomorphic liposarcoma, are considered benign entities and are mainly described in the older adult population. This article will review the currently known morphological, immunohistochemical, and molecular features of this heterogeneous group of mesenchymal tumors with an emphasis on differential diagnosis.

## 1. Introduction

Tumorigenesis has been a major research focus in the last few decades. A plethora of different genetic, epigenetic, and microenvironmental factors influencing tumor development have been elucidated [1]. This research gave rise to many targetable mechanisms that are now used in daily oncology practice. Characteristic gene mutations, fusions, and copy number variations have also been major points of interest in diagnostic pathology. This is especially true for soft tissue tumor pathology, where many previously described tumor entities, based on morphology, have been shown to harbor distinguishing molecular features. Some examples include *EWSR1* gene rearrangement in Ewing sarcoma [2], *MDM2* gene amplification in well-differentiated and dedifferentiated liposarcoma [3], and *c-KIT* gene mutations in gastrointestinal stromal tumor (GIST) [4]. Moreover, an increasing number of the more recently described tumor entities are defined, for a significant part, based on their specific driving genetic mechanism, supporting their separate classification. An illustration of the latter is the introduction of the ‘NTRK-rearranged spindle cell neoplasms’ as an emerging separate entity in the 5th edition of the World Health Organization (WHO) Classification of Soft Tissue and Bone Tumors [5].

One of the best-known genes implicated in tumor development and the first ever described tumor suppressor gene is Retinoblastoma 1 (*RB1*), located at chromosome 13q14.2. As the name suggests, its deletion was originally discovered as a causative factor in aggressive and often bilateral pediatric ocular tumors (retinoblastomas). Familial and/or bilateral retinoblastomas most often develop in children harboring germline *RB1* deletions, only after the occurrence of a “second hit”, corresponding to the loss of the second *RB1* allele. This “two hit” hypothesis, required for oncogenesis, was first proposed by Knudson (1971) based on his observations of retinoblastoma patients [6]. Only years later was it genetically proven that it is homozygous *RB1* deletion that initiates oncogenesis [7].

The *RB1* transcript (RBp) is a chromatin-associated protein of which the primary function is to negatively regulate transcription factor E2F and control the cell cycle. Throughout the cell cycle, RBp is progressively phosphorylated, and thus inactivated, by cyclin-dependent kinases [8]. Although an abundance of literature exploring RBp function exists, the full extent of the role that RBp plays in the cell is still not fully understood [9]. In recent decades, the role of *RB1* loss and the inactivation or dysregulation of RBp function by RB-associated proteins have been implicated in many cancer types, e.g., melanomas, osteosarcomas, small-cell lung cancer, and several mesenchymal tumors [10]. 

In contrast to the typical retinoblastoma sequence, in which all cells of the body harbor a heterozygous germline *RB1* deletion and where biallelic *RB1* loss triggers oncogenesis, most tumor types characterized by *RB1* deletion acquire this by somatic means. As this *RB1* loss occurs in a “normal” genetic background, it is heterozygous. Therefore, several other concurrent tumor suppressor gene inactivations and increased activity of oncogenes, to support tumorigenesis, are usually present in these neoplasms.

Mesenchymal tumors with *RB1* loss represent a heterogeneous family of neoplasms with different morphologic and clinical features, all showing a high incidence of *RB1* deletions. This strongly suggests that *RB1* deletion is an important molecular driver event in the development of these neoplasms, as it is conserved throughout this tumor family. All of these tumors are either adipocytic in nature or can have a lipomatous component. As adipocytic tumors are frequently encountered in daily practice, accounting for about half of all soft tissue neoplasms [5], *RB1*-deleted mesenchymal tumors are often considered in the differential diagnosis. Correct diagnosis is crucial for appropriate medical treatment and prognostication of this heterogeneous group of mesenchymal neoplasms [9]. 

This article will provide a contemporary review on the histopathologic and clinical features of this group of soft tissue tumors with *RB1* loss (spindle cell and pleomorphic lipoma, atypical spindle cell/pleomorphic lipomatous tumor, pleomorphic liposarcoma, myofibroblastoma, cellular angiofibroma, acral fibromyxoma) and their differential diagnoses.

*RB1* loss has also been reported in other mesenchymal neoplasms, such as well-differentiated liposarcomas, myxofibrosarcomas, and undifferentiated pleomorphic sarcomas [11,12,13]. As *RB1* loss is not considered the main driver or the diagnostic feature in these tumors, they will not be discussed in this review.

## 2. Review

### 2.1. The Broad Clinicopathologic Spectrum of RB1-Deleted Mesenchymal Neoplasms

*RB1*-deleted soft tissue tumors are most-commonly seen in the older adult population, usually in patients older than 50 years of age. They can be encountered in a wide variety of anatomical sites. Information concerning the anatomic location of an *RB1*-deleted soft tissue tumor may be crucial in making the correct diagnosis, especially as some of these tumors occur almost exclusively in particular anatomic regions. This is the case, for example, with acral fibromyxoma and cellular angiofibroma, which typically occur in sub (peri-)ungual and inguinal locations, respectively.

Figure 1 illustrates the most frequent anatomical distributions of the *RB1*-deleted soft tissue tumors discussed in this review.

Table 1 gives an overview of the different clinical, pathologic, and molecular characteristics of the heterogenous group of *RB1*-deleted soft tissue tumors. 

#### 2.1.1. Spindle Cell Lipoma and Pleomorphic Lipoma

Spindle cell lipoma (SCL) and pleomorphic lipoma (PL) represent a continuum of benign adipocytic tumors, mainly (80% of cases) arising within the subcutis of the neck, shoulder and back of men aged 45–80 years [14]. These lesions are relatively rare, accounting for approximately 1.5% of all adipocytic neoplasms. They typically present as solitary, slow-growing, and mobile masses that measure <5 cm [15,16]. Less than 10% of these tumors occur in females. Cases in women are more likely to arise outside the typical locations, including the face, scalp, oral cavity, upper and lower limbs, or trunk [15,16,17]. Occasionally, intramuscular or dermal-based lesions can be encountered [18,19,20]. Multiple SCL/PLs occurring simultaneously have been described in the literature and these cases may be familial [21].

Local recurrence is rare, even with incomplete excision [22,23].

Microscopically (Figure 2), SPLs are characterized by variable amounts of mature adipocytes, bland spindle cells, and ropey collagen. Mast cells are often prominent. Mitoses are rare and necrosis is absent. Some tumors show a more prominent vascular network with small- to medium-sized, thick-walled, hyalinized vessels. In PLs, as the name already suggests, pleomorphic and multinucleated floretlike giant cells are encountered in addition to the aforementioned morphologic findings. These components are usually seen in a fibromyxoid background [15,16,22,23]. Lipoblasts are present in up to 50% of cases [24]. Diverse morphologic variants of SCL/PL have been described, which may complicate the differential diagnosis. These include fat-poor, myxoid, (pseudo)angiomatous, and plexiform variants. Cartilaginous or osseous metaplasia and extramedullary hematopoiesis have rarely been reported [16].

Immunohistochemical studies (Figure 2) show diffuse CD34 staining in the spindle, pleomorphic, and floretlike giant cells, as well as loss of Rb staining in most cases [25]. S100 stains adipocytes and variable amounts of spindle cells [26]. Desmin is usually negative [16]. Molecular tests, such as fluorescence in situ hybridization (FISH), show heterozygous 13q14 deletion, including the *RB1* gene.

Differential diagnosis includes conventional lipoma, atypical lipomatous tumor (ALT), atypical spindle cell/pleomorphic lipomatous tumor (ASPLT), cellular angiofibroma, and myofibroblastoma.

Although of little clinical significance, as both tumors are considered benign, conventional lipomas and adipocyte-rich SCL/PL can show some morphologic overlap. These can be differentiated by CD34 immunohistochemical staining, as conventional lipomas are usually negative. Rb immunohistochemistry or molecular tests can also identify Rb loss or *RB1* gene deletion, which are not observed in conventional lipomas. ALT can be excluded on similar grounds, as well as with the use of immunohistochemistry or molecular tests to exclude MDM2 overexpression or MDM2 gene amplification, which are typically seen in ALT. Moreover, atypical spindle cells, which can be seen in ALT, are always absent in SCL/PL.

In contrast with ASPLT, SCL/PL lacks pleomorphic lipoblasts, atypical spindle cells, and “bizarre” pleomorphic stromal/multinucleated cells [27,28]. While infiltrative growth can rarely be seen in SCL/PL [15], it remains an unusual finding that should prompt consideration for an intermediate or malignant neoplasm [27].

Cellular angiofibroma and myofibroblastoma show a preferred localization and a morphology different from SCL/PL, as discussed below. Immunohistochemical studies can be useful in this differential diagnosis, as both cellular angiofibroma and myofibroblastoma usually express desmin and progesterone, an immunohistochemical profile seldom seen in SCL/PL.

Other mesenchymal, nonlipogenic CD34+ neoplasms could also be considered in the differential diagnosis, such as solitary fibrous tumor (SFT) and dermatofibrosarcoma protuberans (DFSP). In difficult cases, these entities can be distinguished from SCL/PL by ancillary tests such as nuclear STAT6 immunoreactivity in SFT and detection of a *COL1A1-PDGFB* fusion in DFSP.

Although pleomorphic liposarcoma (PLS) may show a similar immunohistochemical profile to SCL/PL, differential diagnosis with this entity is generally not an issue due to the marked pleomorphism, cytonuclear atypia, and mitotic activity found in PLS.

#### 2.1.2. Atypical Spindle Cell/Pleomorphic Lipomatous Tumor

Atypical spindle cell/pleomorphic lipomatous tumor (ASPLT) is a benign adipocytic tumor which occurs predominantly in middle-aged adults with a peak incidence in the sixth decade of life [28]. There is a slight male predominance. ASPLTs arise in the subcutis slightly more frequently than in deep (subfascial) soft tissues. Similar tumors were first described by Dei Tos et al. [29] as “spindle cell liposarcoma”. Since, they have been referred to in the literature as “atypical spindle cell lipoma” [30], “dysplastic lipoma” [31], “well-differentiated spindle cell liposarcoma” [32], “anisometric cell lipoma” [33], and “atypical spindle cell lipomatous tumor” [34]. These tumors are now incorporated in the new, 5th edition of the WHO Classification of Tumors of Soft Tissue and Bone (2020) as “Atypical Spindle Cell/Pleomorphic Lipomatous Tumor”.

As this is a relatively recently defined entity, research regarding epidemiology, etiology, and biological behavior is still ongoing [35]. From our experience, it is an underrecognized entity.

Although most patients are >30-years-old, lesions have been described in almost every age group (6–87 years). The most common locations are the hand, foot, and thigh, followed by the shoulder, forearm, lower leg, and knee. The head and neck area, genital area, trunk, and back are less-common locations. Rare cases have been reported in the larynx, trachea, mediastinum, retroperitoneum, and appendix. ASPLT manifests as an enlarging or persistent soft tissue mass with a median size of 5–8.5 cm (range: 0.5–28 cm). Tenderness is sometimes present. The tumor is unencapsulated and shows a multinodular growth pattern with ill-defined margins. 

While ASPLT is regarded as a benign entity by the newest WHO classification (2020), it does have a low recurrence rate of 10–15% for incompletely excised lesions. No metastases or dedifferentiation have been documented.

Microscopically (Figure 3), these tumors show variable amounts of collagenous and/or myxoid extracellular matrix wherein different proportions of mature adipocytes, atypical spindle cells, lipoblasts, and scattered “bizarre” pleomorphic (multinucleated) cells are present. The cellularity can vary greatly, showing a spectrum ranging from low cellularity with mild atypia and abundant extracellular matrix (“atypical spindle cell lipoma” morphology), to highly cellular tumors with mild to moderate atypia, easily identifiable lipoblasts, and scant extracellular matrix (“fibrosarcoma-like lipomatous neoplasm” morphology). In general, the adipocytes show mild to moderate nuclear atypia, and focal bi- or multinucleation. The lipoblasts show varying amounts of pleomorphism. Metaplastic changes can be seen with the presence of smooth muscle, cartilaginous, and osseous elements. Mitotic figures are present but mostly scarce. Necrosis is absent. 

Immunohistochemical studies (Figure 4) show variable, but often diffuse expression of CD34, and loss of Rb expression in most cases. p16 expression is often strong, diffuse to patchy. S100 mainly stains the adipocytes and lipoblasts with variable expression in the spindle and pleomorphic cells. Desmin positivity is seen in a subset of cases (approximately 20%) [34,35]. Molecular studies have shown deletions or losses of 13q14, including *RB1* and its flanking genes (*RCBTB2, DLEU1*, and *ITM2B*) in a significant subset of cases. In addition, monosomy 7 has been reported in some cases.

The main differential diagnoses for ASPLT are spindle cell/pleomorphic lipoma (SCL/PL) (see above), atypical lipomatous tumor (ALT), dedifferentiated liposarcoma (DDLS), myxoid liposarcoma (MLS), and pleomorphic liposarcoma (PLS). While it is not always possible to morphologically differentiate ASPLT from these neoplasms, ALT, DDLS, and MLS harbor specific, recurrent genetic changes, which are not seen in ASPLT. ALT and DDLS are characterized by *MDM2/CDK4* amplifications, while MLS is known to have a *FUS/EWSR1-DDIT3* fusion. PLS, on the other hand, demonstrates a more complex genomic profile with numerous chromosomal gains and losses, frequently also including 13q14 (*RB1*). As karyotyping, array comparative genomic hybridization or copy number variation sequencing is often not readily available for these tumors, PLS can pose a differential diagnostic challenge, especially in the setting of an ASPLT at the high-cellularity end of the spectrum. In these cases, adequate biopsy size and sampling are crucial in making the right diagnosis. In general, PLSs can be differentiated from ASPLTs by a higher degree of pleomorphism, high mitotic activity, and tumor necrosis. In addition, the presence of a pleomorphic lipoma-like component demonstrating floretlike giant cells is not present in PL.

#### 2.1.3. Pleomorphic Liposarcoma

Pleomorphic liposarcomas are fast-growing, usually large (>5 cm), high-grade, malignant adipocytic tumors, for the most part occurring in the older adult population [36]. Pediatric cases are exceptionally rare [37]. These sarcomas are most frequently (in two-thirds of cases) encountered in the extremities, favoring the lower limbs. Less usual sites include the retroperitoneum, trunk wall, and spermatic cord. Most tumors are localized in the deep soft tissues, with only a quarter of cases presenting in the subcutis [38,39]. 

The five-year survival rate is about 60% with long term disease-free survival in only 40% of patients after surgery. Metastases and local recurrences are frequent, both occurring in up to 50% of patients. The most common metastatic site is the lung. Large tumor size, deep location, and high mitotic count are poor prognostic factors [38,39].

Histologically (Figure 5), these tumors are poorly circumscribed with infiltrative margins. The morphology is that of an undifferentiated, pleomorphic sarcoma with varying amounts of dispersed pleomorphic lipoblasts. The undifferentiated component most often consists of spindle cells, with 25% of cases showing epithelioid morphology. Giant and multinucleated tumor cells are frequent. In more than half of cases, the spindle cell areas show at least a focal high-grade myxofibrosarcoma-like morphology [38,39]. 

Adequate sampling to identify (pleomorphic) lipoblasts is crucial in making the diagnosis. Immunohistochemical studies usually have little added value. MDM2 immunohistochemistry and *MDM2* gene FISH analysis to exclude a DDLS may be helpful. S100 and adipophilin staining can help to identify the (pleomorphic) lipoblasts [40,41].

While PLSs reliably show hyperploidy and complex chromosomal rearrangements, resembling other pleomorphic sarcomas, *RB1* loss occurs in up to 50% of cases (Figure 5). Other recurrent genetic abnormalities, albeit less frequent (10–20%), include *NF1* and *TP53* mutations [42,43]. 

Differential diagnosis includes poorly differentiated carcinoma, melanoma, DDLS, ASPLT, pleomorphic sarcoma not otherwise specified (NOS), and myxoid pleomorphic liposarcoma.

Areas with prominent epithelioid morphology can pose a differential diagnostic challenge with nonmesenchymal neoplasms. The main mimickers are clear cell renal cell carcinoma, adrenocortical carcinoma, and melanoma. Although, through adequate sampling combined with clinical information, these tumors often can be differentiated morphologically. Tissue-specific immunohistochemical markers can aid in the diagnosis. Useful immunohistochemical markers to differentiate PLS from renal cell carcinoma and adrenocortical carcinoma are PAX8 and inhibin, melan A and SF1, respectively. A combination of melanocytic markers (S100, SOX10, melan A, HMB-45, PN-L2) can help to differentiate PLS from melanoma.

As mentioned above, MDM2 immunohistochemistry and *MDM2* FISH analysis can be used to rule out DDLS. 

The most reliable way to differentiate this tumor from pleomorphic sarcoma NOS or high-grade myxofibrosarcoma is the identification of true (pleomorphic) lipoblasts. 

ASPLT with prominent pleomorphic lipoblasts may mimic pleomorphic liposarcoma but lacks the high-grade sarcomatous background (see above). Once again, as for the differential diagnosis with high-grade sarcomas, thorough sampling will aid in the diagnosis—in this case, showing more differentiated spindle cell/pleomorphic lipoma-like areas in ASPLT.

Myxoid pleomorphic liposarcoma is an exceedingly rare and aggressive adipocytic tumor, generally observed in a younger population compared to pleomorphic liposarcoma [37]. This tumor occurs in adolescents and young adults, showing similar morphology to myxoid liposarcoma, but with prominent pleomorphism and pleomorphic lipoblasts. These tumors lack the characteristic *DDIT3* rearrangements seen in myxoid liposarcomas. Due to its rarity, molecular characteristics of these tumors have yet to be fully explored. The few available case reports indicate that *RB1* deletion, as well as other large chromosomal gains and losses, seem to play a role in its pathogenesis. In contrast to pleomorphic liposarcomas, gene losses are more abundant in myxoid pleomorphic liposarcomas, but gene amplifications seem to be lacking [44,45].

#### 2.1.4. Myofibroblastoma (of Soft Tissue)

Myofibroblastomas of soft tissue are benign, slow-growing tumors, comparable to myofibroblastomas of the breast [46]. They most commonly (50%) occur in the inguinal and groin area. The median size is 6.5 cm, but some tumors can grow as large as 22 cm. Often, a lump is present for a long time before it becomes symptomatic. These lesions mostly occur in adults in their fifth and sixth decades and are twice more common in males. While typically localized in the inguinal/groin region (including in the vulva/vagina, perineum, and scrotum), they can be encountered in most anatomical subcutaneous sites. Rare examples of deep-seated, intramuscular as well as abdominal, retroperitoneal, or even visceral cases are described in literature. 

Macroscopically, myofibroblastomas are generally well-circumscribed, mobile, and rubbery to gelatinous. Local excision is curative and local recurrences are very exceptional.

Histologically (Figure 6), myofibroblastoma is a well-circumscribed, monotonous, spindle cell tumor with variable cellularity, fascicular growth, and variable amounts of intermixed mature adipocytes. Infiltrative growth is sometimes seen in intramuscular lesions. Rarely, the spindle cells can show prominent schwannoma-like palisading. A spectrum of cellular spindle cell to hypocellular, hyalinized, and even predominantly adipocytic morphologies are described. Mast cells are generally present. Blood vessels are low in density, sometimes hyalinized, and may show a perivascular lymphocytic infiltrate. Ropey collagen is occasionally observed. The spindle cells are usually short with stubby nuclei and eosinophilic to amphophilic cytoplasm. Areas with degenerative-type atypia (enlarged hyperchromatic nuclei and multinucleation) and more epithelioid cell morphology can be present [47].

The classical immunohistochemical profile shows diffuse coexpression of desmin and CD34 (Figure 6). Rarely (<10%), cases are negative for one or both markers. A third of cases also show SMA positivity [47].

Genetically, these tumors consistently show 13q14 loss, including the *RB1* gene, which can be demonstrated by loss of nuclear Rb immunohistochemical staining in 90% of cases [25,47,48].

Due to its varying morphology, myofibroblastomas can have a broad differential diagnosis. The main ones include other benign tumors with *RB1* deletion such as spindle cell lipoma (SCL) and cellular angiofibroma (CAF). SCL and myofibroblastoma show an overlapping morphology and immunohistochemical profile (loss of Rb expression and diffuse CD34 expression). In general, most myofibroblastomas show a less-prominent fat component, mainly consisting of spindle cells with an often hyalinized stromal background. Moreover, desmin immunohistochemistry can help in most cases to differentiate between these tumor entities. Given the overlapping clinical presentation (predilection for inguinal locations) and similar spindle cell morphology, the differential diagnosis with CAF can be challenging. However, the characteristic prominent hyalinized vessels in CAF, as well as immunohistochemical profile (variable CD34 staining and absent or at most focal desmin positivity in cellular angiofibroma), can help to distinguish these entities from each other. As SCL, myofibroblastoma and CAF are closely related, with similar biological behavior, separating these three benign entities is more of an academic endeavor.

Other spindle cell tumors that may resemble myofibroblastoma include (long existing) nodular fasciitis, soft tissue perineurioma, solitary fibrous tumor, low-grade malignant peripheral nerve sheath tumor, spindle cell predominant ASPLT, and dermatofibrosarcoma protuberans. Although, these entities rarely enter the differential diagnosis on resection specimens.

#### 2.1.5. Cellular angiofibroma

Cellular angiofibroma (CAF) is a benign, usually small, and slow-growing tumor arising in the inguinoscrotal region or the vulva, although some tumors, especially in men, can grow up to 25 cm [49]. They occur in the adult population. Affected men are usually older (7th decade) than women (5th decade). CAFs are clinically most often confused with benign vulvovaginal and penoscrotal cysts, lipomas, and hernias [50]. Recurrence after excision is exceedingly rare, even in cases with atypia or sarcomatous change [51,52].

On gross examination, these tumors are firm, well-circumscribed, rarely lobulated lesions, with a white–grey appearance. A pseudocapsule can sometimes be observed. Rarely, infiltration into adjacent musculature does occur.

Histology (Figure 7) shows a highly vascular, spindle cell lesion with a variable amount of edematous to fibrous stroma. Myxoid change is more frequent in men than women. The vessels are small- to medium-sized, thick-walled, and rounded, ectatic, or branching. The spindle cell areas are moderately to highly cellular, composed of short, uniform, elongated cells that may show fascicular growth or nuclear palisading. The nuclei are oval or more elongated with inconspicuous nucleoli. Mitotic figures are generally scarce. Small groups of adipocytes are seen in half of the cases. Perivascular lymphocytic infiltrates and mast cells are often present [50,51,52]. Scattered pleomorphic cells, pleomorphic lipoblasts, and even diffuse sarcomatous changes have been reported. The findings do not seem to impact the prognosis [53,54]. If present, sarcomatous or pleomorphic areas show no signs of necrosis or hemorrhage, and mitotic figures are rare.

Immunohistochemical studies of CAF (Figure 7) characteristically show estrogen and progesterone expression. Variable CD34 expression is not unusual (30–60% of cases). Variable expression of SMA and desmin is seen in a minority of cases [50,52].

*RB1* gene deletion seems to be an important driving genetic mechanism in CAF. This can also be illustrated immunohistochemically, by loss of Rb staining, in most cases [25]. 

The main differential diagnosis includes other benign tumors with *RB1* deletion such as myofibroblastoma and SCL, as discussed above.

Angiomyofibroblastoma (AMFB) shows a similar appearance, but, in addition, demonstrates numerous thin-walled, capillary vessels and is often encapsulated with a thin fibrous capsule. Desmin positivity and lack of Rb loss can reliably differentiate AMFB from CAF. It is also important to exclude aggressive angiomyxoma, as it greatly affects patient management. These lesions occur in younger women in their reproductive years, are paucicellular in comparison, and have infiltrative margins. Prominent myxoid stroma and “spinning off” of spindle cells around vessels are characteristic. In contrast with CAF, aggressive angiomyxomas show diffuse desmin immunoreactivity. 

Due to the high cellularity; CD34 reactivity; and prominent ectatic, branching vessels in some cases of CAF, SFT can also enter the differential diagnosis, especially in small biopsies. STAT6 immunohistochemistry, which is a highly sensitive and specific diagnostic marker for SFT, can be useful in such circumstances.

#### 2.1.6. Acral Fibromyxoma

Acral fibromyxoma, also referred to as superficial acral fibromyxoma or digital fibromyxoma, usually presents as a slowly enlarging and painful subungual or periungual lump [55]. Rare cases in other sites, such as the distal extremities and the thighs, have been documented. These lesions most often occur in middle-aged adults, around 50 years of age. There is a slight predilection for men. They can present with nail deformity and erosion of underlying bone, sometimes seen as a lytic bone lesion on radiography. Local recurrence is seen in up to 24% of cases when incomplete resection is performed. Re-excision is generally curative. Recurrent lesions have a nondestructive growth. Metastases are not described [56,57].

Macroscopically, acral fibromyxomas present as dome-shaped, polypoid, or verrucous lesions, varying in size, usually between 1–2 cm. On cut-sections, they appear lobulated with ill-defined borders. 

Histology (Figure 8) shows a vaguely lobulated, intradermal, or subcutaneous lesion, sometimes with an infiltrative growth pattern. Tumors with polypoid appearance show a surrounding epidermal collarette. Acral fibromyxomas are composed of bland-appearing spindled, ovoid, or stellate cells with slightly eosinophilic cytoplasm in a variable myxoid stroma with some more collagenous areas. Predominantly collagenous or myxoid cases are described. The spindle cells grow haphazardly, in a loose storiform or focally fascicular pattern. Scattered, multinucleated stromal cells and mast cells are regularly present. Degenerative atypia and very scarce mitoses can be seen. Necrosis or cytonuclear pleomorphism are absent. 

Immunohistochemical studies (Figure 8) show diffuse CD34 and variable SMA and EMA expression [56,57].

The primary genetic driving mechanism is *RB1* deletion, which can often be detected immunohistochemically (loss of Rb expression) [58].

Given its specific clinical presentation, diagnosis of acral fibromyxoma is often straightforward. Superficial angiomyxoma can have a very similar myxoid morphology to acral fibromyxoma, albeit lacking the alternating fibrous areas. Acral fibromyxoma does not typically show any interlobular septation, entrapment of adnexal structures, or admixed neutrophils, as seen in superficial angiomyxomas. No association with Carney complex has been documented for acral fibromyxomas [56].

The most clinically significant differential diagnosis is that with low-grade fibromyxoid sarcoma (LGFMS), which shows a similar, alternating myxoid and fibrous histological appearance with bland spindle cells. LGFMS, however, shows a more organized, whirling growth pattern and mostly arises in the deep subcutaneous tissues. In challenging cases, the lack of CD34 expression and diffuse MUC4 staining in LGFMS can guide this diagnosis. Molecular testing for *FUS-CREB3L2/CREB3L1 or EWSR1-CREB3L1* fusions can be of additional value to confirm the diagnosis of LGFMS in selected cases.

Although dermatofibrosarcoma protuberans (DFSP) also strongly stains for CD34 and can sometimes have a prominent myxoid change, differential diagnosis is usually straightforward due to its plaquelike macroscopy, storiform growth pattern, and monotonous appearance. This differs from the more lobulated architecture seen in acral fibromyxomas. Clinical information, as well as molecular analysis (*COL1A1-PDGFB* fusion in DFSP), can help to differentiate DFSP and acral fibromyxoma on smaller biopsy specimens. However, it is crucial to note that the translocation forming the *COL1A1-PDGFB* fusion protein is not widely observed in myxoid DFSPs [59]. Therefore, the absence of the translocation does not definitively rule out DFSP.

Dermal spindle cell lesions, like dermatofibroma and neurofibroma, can sometimes show myxoid change. Neurofibromas show a more uniform cytomorphology, absence of fascicular growth, and diffuse S100 positivity, in contrast to acral fibromyxoma. Dermatofibromas are usually more superficial in comparison to acral fibromyxoma. Due to their localization, clinical appearance, and component of bland spindle cells, acquired (acral) fibrokeratoma and periungual/subungual fibroma may enter the differential diagnosis. Acquired (acral) fibrokeratoma and periungual/subungual fibroma can morphologically be differentiated from acral fibromyxomas by superficial growth, very low cellularity, and less vascular proliferation. No association with tuberous sclerosis, as seen in periungual/subungual fibromas, has been observed for acral fibromyxoma [56]. 

Acral fibromyxoma with prominent myxoid features is more cellular with a higher vascularity than other benign myxoid lesions, such as solitary cutaneous myxoma, dermal mucinosis, and intramuscular myxoma. Cellular myxoma of soft tissues does shows a higher cellularity and CD34 staining but is composed of more slender spindle cells with a completely disorganized growth pattern compared to acral fibromyxoma. Moreover, cellular myxomas usually occur in the deep soft tissue. Acral fibromyxomas do not contain *GNAS1* mutation, as seen in intramuscular and cellular myxomas.

## 3. Conclusions

This contemporary review on ‘the rapidly expanding group of *RB1*-deleted soft tissue tumors’ highlights the clinicopathologic features of this heterogeneous group of mesenchymal neoplasms. Diagnosis of these entities can be challenging due to their rarity, as well as their clinical and histologic overlap. Adequate sampling and the awareness of the different clinical, histologic, immunophenotypic, and molecular features are important to navigate the differential diagnosis and to make the correct diagnosis.

## Figures and Tables

**Figure 1 diagnostics-11-00430-f001:**
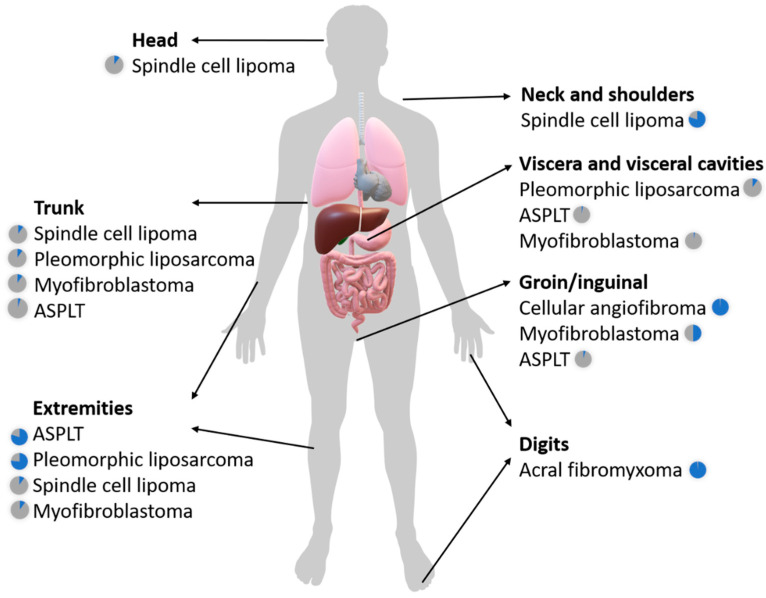
Anatomical distribution of the different *RB1*-deleted mesenchymal neoplasms. Pie charts next to the tumor names represent the frequency of a given tumor occurring in that specific location (blue pie segment). ASPLT—Atypical spindle cell/pleomorphic lipomatous tumor.

**Figure 2 diagnostics-11-00430-f002:**
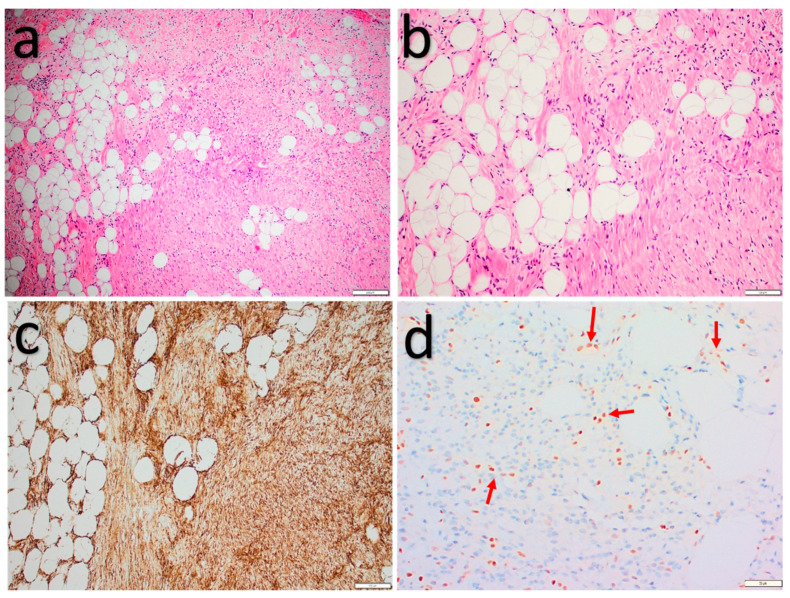
Spindle cell lipoma. (**a**,**b**) Bland spindle cells set in a fibromyxoid stroma mixed with mature adipocytes. (**c**) Strong, diffuse, cytoplasmatic CD34 expression. (**d**) Loss of nuclear Rb expression in the spindle cells and intact nuclear Rb expression in the endothelial cells (red arrows).

**Figure 3 diagnostics-11-00430-f003:**
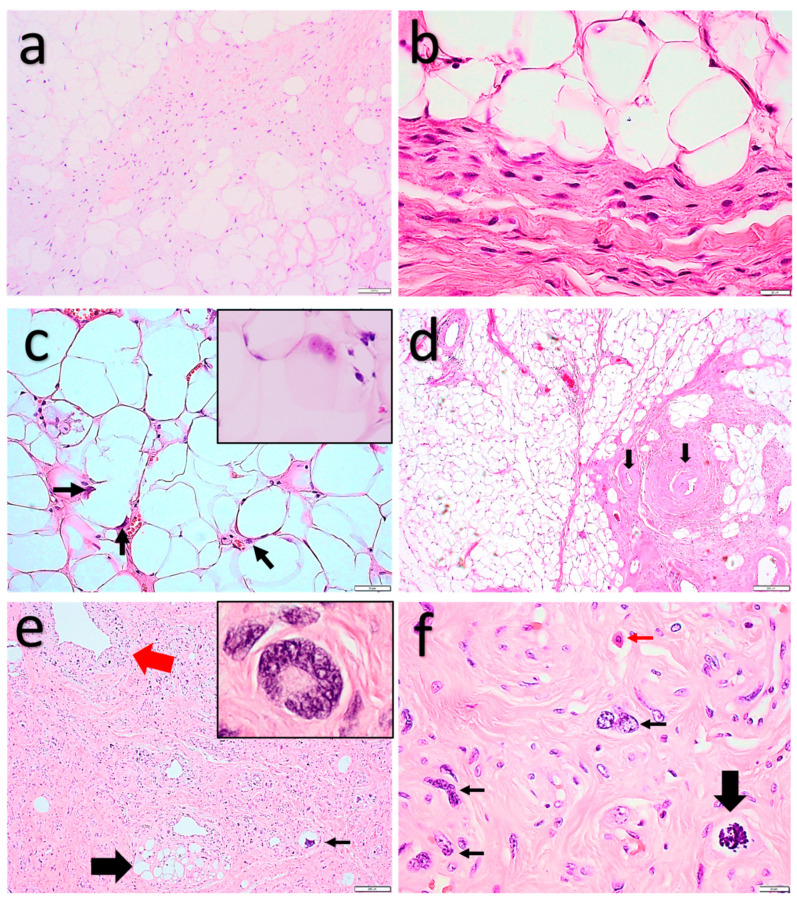
Atypical spindle cell/pleomorphic lipomatous tumor. (**a**,**b**) Low-cellularity end of the spectrum: mature adipocytes and fibrous septa with slightly increased cellularity. In (**a**), a more myxoid stroma can be seen. (**b**) Low-cellularity end of the spectrum: spindle cells with mild cytonuclear atypia, adipocytes with some variation in shape and size. (**c**) Adipocytes with mild cytonuclear atypia (arrows) and binucleation (inset) (adipocytic-rich variant, ‘dysplastic-lipoma’-like morphology). (**d**) Prominent hyalinized vessels. (**e**) High-cellularity end of the spectrum: focal adipocytic component showing adipocytes with variation in size and shape (large black arrow), cellular adipocytic-poor tumor component with perivascular condensation of pleomorphic tumor cells (large red arrow), pleomorphic lipoblasts (small black arrow), and floretlike multinucleated cells (inset). (**f**) High-cellularity end of the spectrum: scattered pleomorphic and multinucleated “bizarre”cells (small black arrows), a mitotic figure (large black arrow), and mast cell (red arrow). (**f**) High-cellularity end of the spectrum: scattered pleomorphic and multinucleated “bizarre”cells (small black arrows), a mitotic figure (large black arrow), and mast cell (small red arrow).

**Figure 4 diagnostics-11-00430-f004:**
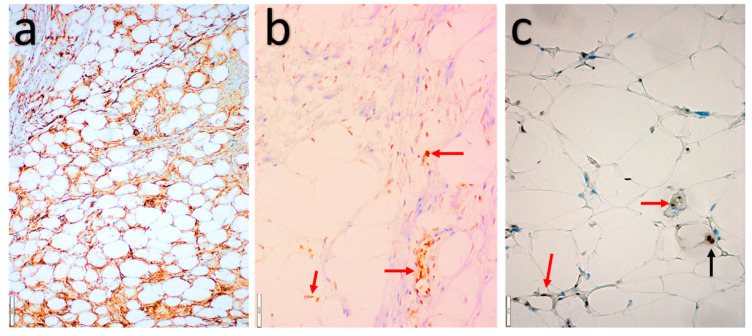
Atypical spindle cell/pleomorphic lipomatous tumor. (**a**) Strong, diffuse, cytoplasmatic CD34 expression. (**b**,**c**) Loss of nuclear Rb expression in the adipocytes and spindle cells, and intact nuclear Rb expression in the endothelial cells (red arrows) and macrophage (black arrow).

**Figure 5 diagnostics-11-00430-f005:**
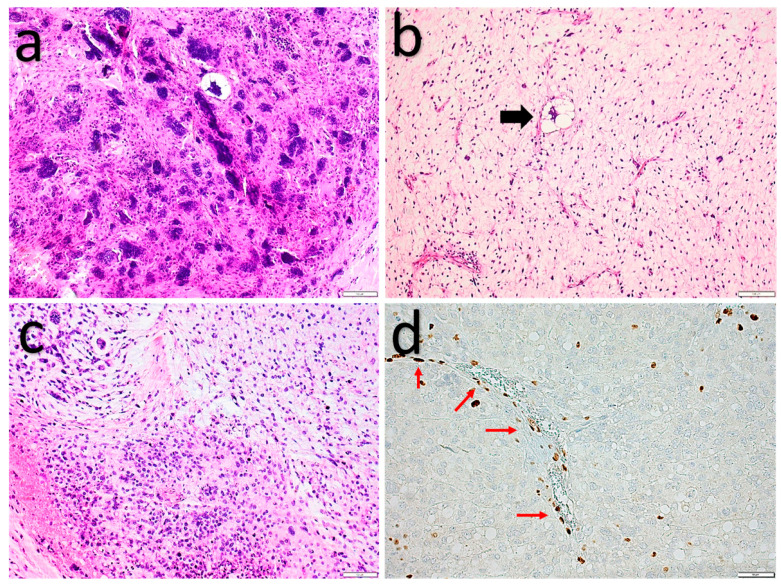
Pleomorphic liposarcoma. (**a**) High-grade pleomorphic sarcoma morphology with pleomorphic lipoblasts (arrow). (**b**) Myxofibrosarcoma-like morphology with pleomorphic lipoblast (arrow). (**c**) High-grade epithelioid sarcoma morphology. (**d**) Loss of nuclear Rb expression in the tumor cells and intact nuclear Rb expression in the endothelial cells (red arrows).

**Figure 6 diagnostics-11-00430-f006:**
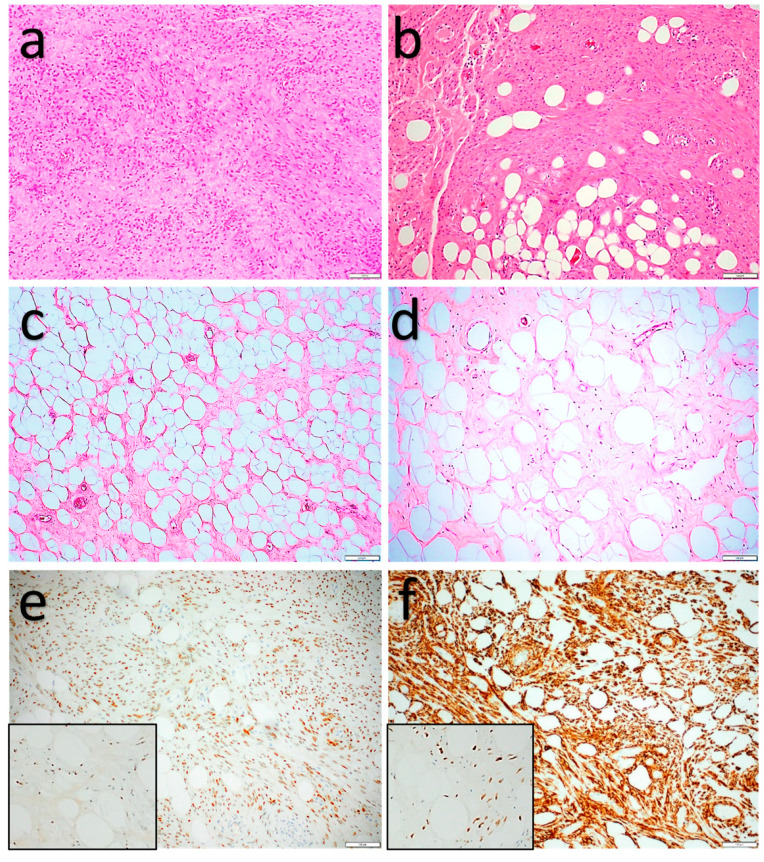
Myofibroblastoma. (**a**) Spindle cell tumor with fascicular growth showing short fascicles and collagenous background. (**b**) Bland spindle cells with eosinophilic cytoplasm and intermixed mature adipocytes. (**c**) Adipocytic-rich variant of myofibroblastoma with a dominant lipomatous component. (**d**) Adipocytic-rich variant: mature adipocytes and bland, haphazardly orientated spindle cells in a fibrous stroma. (**e**) Strong, diffuse nuclear progesterone expression. Inlet showing an adipocytic-rich variant. (**f**) Strong, diffuse cytoplasmatic desmin expression. Inlet showing an adipocytic-rich variant.

**Figure 7 diagnostics-11-00430-f007:**
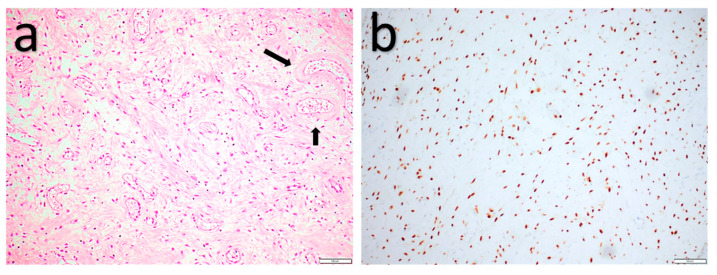
Cellular angiofibroma. (**a**) Bland spindle cells in a variably fibrous and hyalinized stroma with numerous hyalinized, small- to medium-sized vessels (arrows). (**b**) Strong, diffuse nuclear progesterone positivity.

**Figure 8 diagnostics-11-00430-f008:**
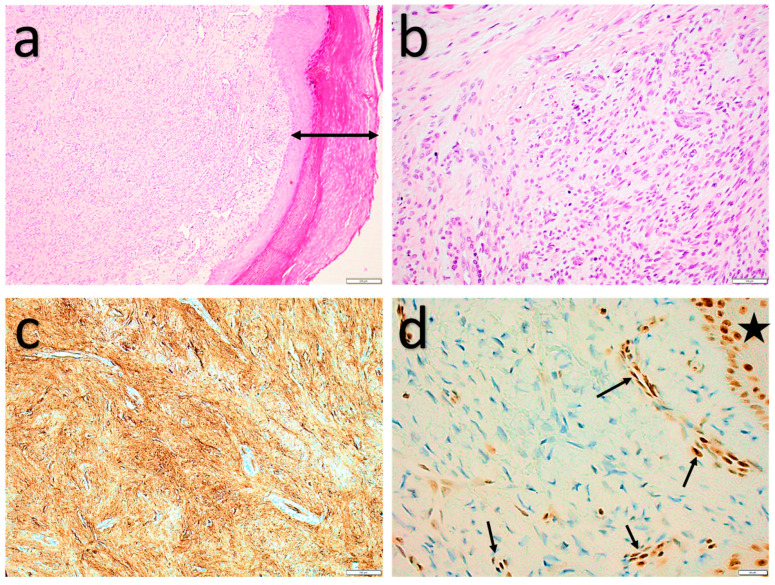
Acral fibromyxoma. (**a**) Superficial, dermal, bland fibromyxoid spindle cell lesion with hyperplasia of the overlying epidermis (double-headed arrow). (**b**) Bland spindled cells set in a fibrous and fibromyxoid stroma. (**c**) Strong, diffuse, cytoplasmatic CD34 expression. (**d**) Loss of nuclear Rb expression in the spindle cells and intact nuclear Rb expression in the endothelial cells (arrows) and epidermis (star).

**Table 1 diagnostics-11-00430-t001:** Summary of the clinical, histopathological and molecular characteristics of *RB1*-Deleted Mesenchymal Neoplasms.

Tumor Types	SCL/PL	ASPLT	PLS	MFB	CAF	AFM
**Sex ratio (M:F)**	10:1	1.5–3:1	1:1	2:1	1:1 (men older than women)	1.2–2:1
**Recurrence**	Very rare local recurrence	10–15% local recurrence if incompletely excised	50% local recurrence 50% metastases 60% 5 y OS	Very rare local recurrence	Very rare local recurrence, even with sarcomatous change	1/4th of lesion locally recur after incomplete excision; re-excision is curative.
**Macroscopic appearance**	Circumscribed, encapsulated Slow-growing Size < 5 cm	Vaguely lobular, unencapsulated, ill-defined margins Gradually enlarging Median size 5–8.5 cm (up to 28 cm)	Poorly circumscribed, infiltrative Fast growing Large (usually >5 cm)	Well circumscribed, nodular Rarely infiltrative Very slow-growing Median size 6.5 cm (up to 22 cm)	Well circumscribed, nodular, sometimes with pseudocapsule Rarely infiltrative Slow-growing Median size 2–3 cm in women and 6–7 cm in men (up to 25 cm)	Polypoid/verrucous, lobulated, sometimes infiltrative Slow-growing Recurrent lesions are always well-circumscribed. Size 1–2 cm
**Histology** **Background**	Variable fibromyxoid stroma, ropey collagen, mast cells	Variable fibromyxoid stroma, ropey collagen, mast cells, perivascular lymphocytic infiltrate	Highly cellular with scant fibromyxoid stroma	Fibrous stroma with variable hyalinized areas, ropey collagen, mast cells, perivascular lymphocytic infiltrate	Variably edematous and fibrous stroma, sometimes myxoid change, mast cells, perivascular lymphocytic infiltrate	Variably myxoid stroma with fibrous areas, mast cells
**Cellularity**	Low	Highly variable	High	Low to moderate	Moderate to high	Low
**Cell type and growth pattern**	Bland spindle cells, mature adipocytes, lipoblasts, scattered bizarre cells	Variably atypical spindle cells, atypical adipocytes, (pleomorphic) lipoblasts, (bizarre) floretlike giant cells	Highly atypical spindle cells, highly atypical epithelioid cells (25%), pleomorphic lipoblast (var), giant and multinucleated tumor cells	Bland spindle cells with fascicular growth, intermixed mature adipocytes, rare degenerative-type atypia, rare epithelioid cells	Short, bland spindle cells, sometimes nuclear palisading, and fascicular growth Rarely pleomorphic spindle cells, lipoblasts, and sarcomatous change	Bland spindled, ovoid or stellate cells, vague fascicular or storiform growth, dispersed small multinucleated stromal cells
**Mitoses/necrosis**	Very rare mitoses No necrosis	Often present but scarce mitoses No necrosis	High mitotic count Frequent necrosis	Very rare mitoses No necrosis	Scarce mitoses, rarely increased No necrosis	Very rare mitoses No necrosis
**Vasculature**	Low density, hyalinized	Low to moderate density, hyalinized	High vascularity, sometimes chicken wire-like vasculature	Low density, sometimes hyalinized	High vascularity, variable diameter, prominent hyalinization	Low density, inconspicuous
**Heterologous differentiation**	Cartilaginous, osseous, EMH	Smooth muscle, cartilaginous, osseous	Highly variable	Absent	Absent	Cartilaginous
**IHC**	CD34(+) S100(+) in adipocytes Desmin(-) PR rare Rb loss	CD34(+) S100(+) in adipocytes and S100 (var) in spindle cells Desmin (var) Rb loss (60–100%)	CD34 (-/var) S100 (+) in lipoblasts Desmin (-/var) Rb loss (±50%)	CD34 (+) S100 (-), except adipocytes Desmin (+) SMA (30%) Rb loss (90%)	CD34 (var) S100 (-) SMA/Desmin (rare) ER/PR (+) Rb loss	CD34 (+) S100 (-) Desmin (-) SMA (var) EMA (var) Rb loss (>90%)
**Molecular**	Heterozygous 13q14 deletion	Heterozygous 13q14 deletion (60-70%)	CNVs (loss > gain) Rearrangements Heterozygous 13q14 deletion (50%)	Heterozygous 13q14 deletion	Heterozygous 13q14 deletion	Heterozygous 13q14 deletion

IHC—Immunohistochemistry; SCL/PL—spindle cell lipoma/pleomorphic lipoma; ASPLT—atypical spindle cell/pleomorphic lipomatous tumor; PLS—pleomorphic liposarcoma; CAF—cellular angiofibroma; AFM—acral fibromyxoma; MFB—myofibroblastoma (of soft tissue); EMH—extramedullary hematopoiesis; M:F—male to female ratio; 5 y OS—five-year overall survival; var—variable; ER—estrogen receptor; PR—progesterone receptor; CNV—copy number variant.

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
