# Peer review of "The Rapidly Expanding Group of RB1-Deleted Soft Tissue Tumors: An Updated Review"

_diagnostics, 2021, doi:10.3390/diagnostics11030430_

Round 1

Reviewer 1 Report

The authors describe a group of Rb1-deleted soft tissue tumors. The manuscript is well written and Table included summarizes all entities nicely.

Major remarks:

  1. Section 2.1.2. ASPLT. In the literature, several cases with dedifferentiation and poor outcome have been described (on page 6 2. Paragraph authors stated that no dedifferentiation has been documented). Authors should comment on that and their personal experience, as this is somehow confusing to the readers as in the new WHO this entity is included in the benign lipomatous tumors. In addition, on page 6, in the 5. paragraph all “old” names used in the literature are listed. This should come at the beginning of the section to define the entity more clearly and to avoid repetition (about unknown etiology).
  2. Page 8, Figure 3. The description for image 3f is missing, please add the description. The image c, inlet is unclear - it should show bi/multinucleated cell. Please include better figure. Also, in description for figure e authors are mentioning atypical adipocytes - atypical should be removed as this terminology is not used. 
  3. Figure 4. Rb1 staining in a tumor with more spindle cells would be better, instead of alcian blue.
  4. Pleomorphic LSA: We found in our practice that adipophilin stain nicely highlights the lipoblast and helps in differentiating between other entities in the differential diagnosis (like pleomorphic sarcoma NOS). Could authors please comment on that in this section?
  5. Myofibroblastoma: In Figure 6, one image with more typical morphology should be added showing spindle cell proliferation arranged usually in short fascicles with interspersed bands of hyalinized collagen. In (e) the positive reaction in inlet is not clearly visible, a higher magnification should be used.

Minor remarks:

  1. Page 4, 2.1.1, second paragraph, first sentence “SCL/PLs are benign tumors” should be removed, as this was already stated in the first paragraph.
  2. Page 6. Figure 2, images c and d are mixed – I would suggest putting Rb1 staining instead of alcian blue, as alcian blue is not useful for the diagnosis.
  3. Page 11, Fig. 5. a-b) undifferentiated is not necessary and should be removed from the description, as readers could be confused. Image d in the description lacks parenthesis “(d)”
  4. Page 12, 2. Paragraph, “Ropey collagen is often occasionally observed” – “often” should be removed.

Author Response

Dear reviewer,

Thank you for taking your time and thoroughly reading through our review. We appreciate all the remarks you have made, and we implemented them where possible.
The summary of implemented changes can be found below, these are highlighted in red/blue in the main text.

Kind regards,
Sasha Libbrecht

SUMMARY OF REVIEWER REMARKS AND ADJUSTMENTS

REVIEWER 1

Comments and Suggestions for Authors

The authors describe a group of Rb1-deleted soft tissue tumors. The manuscript is well written and Table included summarizes all entities nicely.

Major remarks:

  1. Section 2.1.2. ASPLT. In the literature, several cases with dedifferentiation and poor outcome have been described (on page 6 2. Paragraph authors stated that no dedifferentiation has been documented). Authors should comment on that and their personal experience, as this is somehow confusing to the readers as in the new WHO this entity is included in the benign lipomatous tumors. In addition, on page 6, in the 5. paragraph all “old” names used in the literature are listed. This should come at the beginning of the section to define the entity more clearly and to avoid repetition (about unknown etiology).

  • As far as we know, no clear cases of ASPLT with dedifferentiation have yet been reported in literature. Although we know of some existing cases with presumed dedifferentiation to pleomorphic liposarcoma, they are still part of ongoing research. After discussion with the WHO editorial board, it has been decided to classify ASPLT as a benign entity with a possibility of local recurrence in the most recent WHO bluebook publication. As this is a review, we feel that mentioning the possibility of dedifferentiation without any citing literature is not appropriate.
  • The order of paragraph 5, on page 6, has been changed.

  1. Page 8, Figure 3. The description for image 3f is missing, please add the description. The image c, inlet is unclear - it should show bi/multinucleated cell. Please include better figure. Also, in description for figure e authors are mentioning atypical adipocytes - atypical should be removed as this terminology is not used. 

  • Description for image 3f was added
  • A new picture of a bi-nucleated lipocyte has been added in the inlet of Figure 3, image c.
  • “atypical” was removed from the description in figure 3, image e.

  1. Figure 4. Rb1 staining in a tumor with more spindle cells would be better, instead of alcian blue.

  • Alcian blue stain was removed in Figure 2 and Figure 4, instead a picture of Rb IHC stain was added.

  1. Pleomorphic LSA: We found in our practice that adipophilin stain nicely highlights the lipoblast and helps in differentiating between other entities in the differential diagnosis (like pleomorphic sarcoma NOS). Could authors please comment on that in this section?

  • We have no experience with the adipophilin stain, but we added this (with reference) as an other possible way to highlight lipoblasts.

  1. Myofibroblastoma: In Figure 6, one image with more typical morphology should be added showing spindle cell proliferation arranged usually in short fascicles with interspersed bands of hyalinized collagen. In (e) the positive reaction in inlet is not clearly visible, a higher magnification should be used.

  • A new picture with more typical morphology has been added to figure 6
  • We zoomed in on the photos in the inlets. The positive cells should be clearly visible now.

Minor remarks:

  1. Page 4, 2.1.1, second paragraph, first sentence “SCL/PLs are benign tumors” should be removed, as this was already stated in the first paragraph.

  • Adjustment has been made

  1. Page 6. Figure 2, images c and d are mixed – I would suggest putting Rb1 staining instead of alcian blue, as alcian blue is not useful for the diagnosis.

  • Alcian blue stain was removed in Figure 2 and Figure 4, instead a picture of Rb IHC stain was added.

  1. Page 11, Fig. 5. a-b) undifferentiated is not necessary and should be removed from the description, as readers could be confused. Image d in the description lacks parenthesis “(d)”

  • Adjustment has been made

  1. Page 12, 2. Paragraph, “Ropey collagen is often occasionally observed” – “often” should be removed.

  • Adjustment has been made

Reviewer 2 Report

Comments:

The authors should review and discuss some previous studies as follows:

(1) Schneider-Stock et al (doi: 10.1002/path.1145) and Takahira et al (doi: 10.1038/modpathol.3800447) reported a significance of RB1-deletion during tumor progression in WDLS (well-differentiated liposarcomas).

(2) Li et al (doi: 10.1158/0008-5472) showed that they identified shallow or deep deletions of RB1 in 68% of patients with MFS (myxofibrosarcoma) and/or UPS (undifferentiated pleomorphic sarcoma).

(3) Chen et al (doi: 10.1097/PAS.0b013e31825d532d) demonstrated that nuclear Rb expression was deficient in patients with SCL/PL (100%), CAF (100%), MFB(89%), and lipoma (9%).

Minor concerns:

(4) The tumor types in Table 1 should be arranged by the order in the text; “MFB” are moved into between “PLS” and “CAF.”

Author Response

Dear reviewer,

Thank you for taking your time and thoroughly reading through our review. We appreciate all the remarks you have made, and we implemented them as best as we could. I hope the added parts to the introduction clarify things better.
The summary of implemented changes can be found below, these are highlighted in red/blue in the main text.

Kind regards,
Sasha Libbrecht

SUMMARY OF REVIEWER REMARKS AND ADJUSTMENTS

REVIEWER 3

The authors should review and discuss some previous studies as follows:

(1) Schneider-Stock et al (doi: 10.1002/path.1145) and Takahira et al (doi: 10.1038/modpathol.3800447) reported a significance of RB1-deletion during tumor progression in WDLS (well-differentiated liposarcomas).

(2) Li et al (doi: 10.1158/0008-5472) showed that they identified shallow or deep deletions of RB1 in 68% of patients with MFS (myxofibrosarcoma) and/or UPS (undifferentiated pleomorphic sarcoma).

(3) Chen et al (doi: 10.1097/PAS.0b013e31825d532d) demonstrated that nuclear Rb expression was deficient in patients with SCL/PL (100%), CAF (100%), MFB(89%), and lipoma (9%).

- The 3 first references (Schneider-Stock et al, Takahira et al, Li et al) are now mentioned and discussed in the introduction.

- The paper of Chen et al. is added to the section discussing immunohistochemistry in SCL/PL, CAF and MFB. 

Minor concerns:

(4) The tumor types in Table 1 should be arranged by the order in the text; “MFB” are moved into between “PLS” and “CAF.”

- Order adjusted

Reviewer 3 Report

This review paper by Libbrecht et al summarized RB1-deleted soft tissue tumors. They summarized each soft tissue tumors types and added beautiful tissue/tumor images.

Overall this is well written manuscript and I don't have too much complains. But authors should clarify RB1 deleted tumors is RB1+/- as they wrote in Table 1. RB1 deleted may mislead readers as RB1 homologous mutants by LOH as seen retinoblastoma.

It would be nice to have one section about RB1 and retinoblastoma to explain the molecular mechanisms of RB1 proteins and tumor genesis by expanding more Introduction section. This is fundamental but authors can differentiate and stress the differences from RB1 deletion in soft tissue tumors.

Author Response

Dear reviewer,

Thank you for taking your time and thoroughly reading through our review. Thank you for pointing this out, we added a section to the introduction mentioning RB1 deletions in other mesenchymal tumors.

The summary of implemented changes can be found below, these are highlighted in red/blue in the main text.

Kind regards,
Sasha Libbrecht

SUMMARY OF REVIEWER REMARKS AND ADJUSTMENTS

REVIEWER 2

Comments and Suggestions for Authors

This review paper by Libbrecht et al summarized RB1-deleted soft tissue tumors. They summarized each soft tissue tumors types and added beautiful tissue/tumor images.

Overall this is well written manuscript and I don't have too much complains. But authors should clarify RB1 deleted tumors is RB1+/- as they wrote in Table 1. RB1 deleted may mislead readers as RB1 homologous mutants by LOH as seen retinoblastoma.

It would be nice to have one section about RB1 and retinoblastoma to explain the molecular mechanisms of RB1 proteins and tumor genesis by expanding more Introduction section. This is fundamental but authors can differentiate and stress the differences from RB1 deletion in soft tissue tumors.

  • The introduction was rewritten to better explain this difference and the role of RB1 in tumorigenesis is now discussed in greater detail.

Round 2

Reviewer 2 Report

The authors have satisfactorily addressed my concerns.